# Efficacy of Shock Wave-Enhanced Emission Photoacoustic Streaming (SWEEPS) in the Removal of Different Combinations of Sealers Used with Two Obturation Techniques: A Micro-CT Study

**DOI:** 10.3390/ma16083273

**Published:** 2023-04-21

**Authors:** Anja Baraba, Marko Rajda, Gorana Baršić, Silvana Jukić Krmek, Damir Šnjarić, Ivana Miletić

**Affiliations:** 1Department of Endodontics and Restorative Dentistry, School of Dental Medicine, University of Zagreb, 10000 Zagreb, Croatia; baraba@sfzg.hr (A.B.); jukic@sfzg.hr (S.J.K.); 2Dental Polyclinic Zagreb, Perkovčeva ul. 3, 10000 Zagreb, Croatia; rajdamarko@gmail.com; 3Department of Quality, Faculty of Mechanical Engineering and Naval Architecture, Ivana Lučića 5, 10002 Zagreb, Croatia; gorana.barsic@fsb.hr; 4Department of Endodontics and Restorative Dentistry, Faculty of Dental Medicine, University of Rijeka, 51000 Rijeka, Croatia; damirsnjaric@gmail.com

**Keywords:** SWEEPS, calcium silicate sealer, single-cone, core-carrier, re-treatment, micro-CT analysis

## Abstract

This study sought to evaluate the efficacy of SWEEPS in the removal of epoxy-resin-based and calcium-silicate-containing endodontic sealer combined with single-cone and carrier-based obturation techniques through a micro-CT analysis. Seventy-six single-rooted extracted human teeth with single root canal were instrumented with Reciproc instruments. Specimens were randomly divided into four groups (n = 19) according to the root canal filling material and obturation technique: (1) AH Plus sealer + Reciproc gutta-percha, (2) TotalFill BC sealer + TotalFill BC Points, (3) AH Plus sealer + Guttafusion obturator, and (4) MTA Fillapex + Guttafusion obturator. All specimens were re-treated one week later using Reciproc instruments. Following re-treatment, root canals were additionally irrigated using the Auto SWEEPS modality. The differences in the root canal filling remnants were analyzed by micro-CT scanning of each tooth after root canal obturation, after re-treatment, and after additional SWEEPS treatment. Statistical analysis was performed using an analysis of variance (*p* < 0.05). The additional treatment with SWEEPS significantly reduced the volume of the root canal filling materials in all experimental groups compared to the removal of root canal filling using only reciprocating instruments (*p* < 0.05). However, the root canal filling was not removed completely from any of the samples. SWEEPS can be used to enhance the removal of both epoxy-resin-based and calcium-silicate-containing sealers, in combination with single-cone and carrier-based obturation techniques.

## 1. Introduction

Endodontic re-treatment can be challenging as root canal filling materials are not easily removed from the endodontic space and their complete removal is often impossible [1]. Unfortunately, remnants of the previous root canal filling may act as a mechanical barrier between the irrigation solution and microbes located in hard-to-reach areas, such as dentinal tubules, lateral canals, and isthmuses, which might explain the low success rate of endodontic re-treatment reported in the literature [1,2,3,4,5,6,7]. Both the chemical composition of root canal sealers and obturation techniques may influence the successful removal of the root canal filling materials. Calcium-silicate-containing endodontic materials are the most recently introduced type of root canal sealer [8]. These sealers set due to a hydration reaction which results in the formation of calcium silicate hydrate gel and portlandite, the latter reacting with the phosphate ions in dentinal fluid and forming hydroxyapatite [8,9]. While this biomineralization improves the adhesion of the sealer to the root canal dentin and promotes successful sealing, it can also make its removal difficult if endodontic re-treatment is required. In order to improve and simplify obturation of the endodontic space, various root canal filling techniques have been developed, such as the single-cone technique with cold gutta-percha and core-carrier systems, which utilizes thermoplasticized gutta-percha. Warm obturation techniques improve the compaction of gutta-percha into root canals and all canal irregularities and allow a deeper penetration of sealers in comparison to cold gutta-percha [10]. Although good penetration of the root canal filling materials into the dentin tubules is required to obtain a tight seal, it may also prevent complete removal of the materials during re-treatment [11].

The effectiveness of debridement during endodontic treatment or re-treatment has been improved by the introduction of different irrigation activation techniques [12]. This approach is based on the idea that by activating irrigating solutions, more thorough disinfection of the root canal system can be achieved [13]. A similar effect can also be achieved with lasers. Laser-activated irrigation (LAI) is based on the generation of intracanal cavitation due to photoacoustic and photomechanical effects [14]. However, potential limitations of LAI include the need for placement of the laser fiber in the root canal a few millimeters short of the apical foramen [12], especially in teeth with a complex endodontic space morphology and severe curvatures of the root canals [15]. In response, specific LAI techniques, including photon-induced photoacoustic streaming (PIPS) and shock wave-enhanced emission photoacoustic streaming (SWEEPS), were introduced to the market. In both PIPS and SWEEPS, the laser fiber is characteristically placed in the pulp chamber rather than inserted into the canal, as pulp chamber placement is less invasive and clinically easier to achieve. Compared to PIPS, SWEEPS technology provides improved cleaning and disinfection of the root canal system due to greater amplification of pressure waves [14].

However, to date, there is limited information on the efficacy of SWEEPS in the removal of different types of root canal sealers used with various obturation techniques. Therefore, the objective of the present study was to investigate the effectiveness of SWEEPS in the removal of epoxy-resin-based and calcium-silicate-containing sealers when used in combination with single-cone and core-carrier techniques. The null hypothesis being tested was that there will be no difference in the effectiveness of SWEEPS in the removal of epoxy-resin-based and calcium-silicate-containing sealers when used in combination with single-cone and core-carrier techniques.

## 2. Materials and Methods

### 2.1. Preparation of Samples

This study was approved by the Ethics committee of the School of Dental Medicine, University of Zagreb, Croatia (No: 05-PA-30-IX-9/2019). In this study, 76 single-rooted extracted human teeth with single canals and intact and mature root apices were selected. The preparation of samples, obturation and removal of root canal filling was performed by a single operator with six years of clinical experience, while the SWEEPS treatment was performed by another operator who specialized in endodontics with clinical experience of more than ten years. Contaminated tissue on the outer specimen surfaces was removed with a hand scaler before any treatment or scanning to maintain aseptic conditions and operator safety. The working length of each root canal was determined by inserting a size #15 K file filler (DentsplyMaillefer, Ballaigues, Switzerland) into the canal until the tip of the file was at the apical foramen and then subtracting 1 mm. To obtain roots with a standardized length of 17 mm, the crowns were sectioned using a water-cooled diamond drill. Root canal instrumentation was performed using a size R25 Reciproc instrument and a VDW Gold endo motor (VDW, Munich, Germany), according to the manufacturer’s instructions. During instrumentation, the canals were irrigated with a 2.5% NaOCl solution using a 27-gauge needle and a 2 mL syringe. The smear layer was removed by rinsing the root canals with 2 mL of 17% EDTA (pH 7.7) for 1 min, followed by a final rinse with saline. The canals were dried using size R25 Reciproc paper points (VDW, Munich, Germany).

### 2.2. Obturation Techniques

All specimens were randomly divided into four experimental groups according to the root canal obturation technique and the type of root canal sealer. The canals were obturated using either a cold single-cone technique (groups 1 and 2) or a warm core-carrier technique (groups 3 and 4). 

In group 1 (n = 19), AH Plus sealer (DeTreyDentsply, Konstanz, Germany) (Table 1) was mixed on a paper pad. The Reciproc R25 gutta-percha cone (VDW, Munich, Germany) was dipped into the sealer and then moved slowly, in an up-and-down motion, until it reached the canal full working length. In group 2 (n = 19), a combination of bioceramic gutta-percha (TotalFill BC Points, FKG, La Chaux de Fonds, Switzerland, 25.06) and bioceramic sealer (TotalFill, FKG, La Chaux de Fonds, Switzerland) (Table 1) was used for the root canal filling. After TotalFill BC sealer was syringed into the canal, TotalFill gutta-percha was placed in the canal up to the working length. The coronal excess of the master cone was cut to the coronal orifice using a flame-heated hand plugger in both experimental groups.

In group 3 (n = 19), the root canal walls were coated with AH Plus sealer using a size #25 reamer (VDW, München, Germany) in a counterclockwise motion. In group 4 (n = 19), MTA Fillapex sealer (Angelus Soluções Odontológicas, Londrina, Brazil) (Table 1) was used to coat the root canal walls. For groups 3 and 4, Guttafusion obturator R25 (VDW, München, Germany) was heated in a special oven while the canal was being coated with the root canal sealer. The heated gutta-percha was then slowly inserted up to the working length without twisting or forcing. The excess material in the canal orifice was then extruded by bending the core-carrier to the right and left until separation took place, and the core material was condensed with a plugger by the same researcher. 

The access cavity in all specimens in the four experimental groups was sealed with glass ionomer cement (Equia Fil, GC, Tokyo, Japan). All specimens were stored in saline at 37 °C for one week to allow sufficient time for the sealer to set.

### 2.3. Removal of the Root Canal Filling

After one week, and complete setting of the sealer, the root canal re-treatments were performed on all specimens (n = 76). For re-treatment, size R25 Reciproc instruments (VDW, Munich, Germany) were used in a VDW Gold endo motor (VDW, Munich, Germany) according to the manufacturer’s instructions, without the use of any solvent. The root canals were rinsed with 2 mL of 2.5% NaOCl solution. Removal of the root canal filling materials from the root canals was considered completed when smooth canal walls were observed and no evident root canal filling material was found on the Reciproc instruments. At the end of the re-treatment procedure, the root canals were rinsed with 2 mL of 17% EDTA (pH 7.7) for 1 min, followed by a final rinse with saline solution. The canals were then dried using size R25 Reciproc paper points (VDW, Munich, Germany).

### 2.4. SWEEPS Treatment

Laser-activated irrigation was performed using an Er:YAG laser (Fotona d.o.o., Ljubljana, Slovenia) with a 2940 nm wavelength, following the auto SWEEPS protocol (50 μs, 15 Hz, 20 mJ, and 0.3 W). The laser was equipped with a dental handpiece (H14; Fotona d.o.o., Ljubljana, Slovenia) and optically coupled with an interchangeable fiber tip (FT). The pulp chamber was reconstructed using thermoplastic materials (Bite compound GC, Tokyo, Japan) and served as a reservoir for the irrigation solution. The tip was submerged in saline and hovered above the orifice in the cervical region rather than being inserted into the canal. Coaxial water and air sprays were deactivated. In all cases, saline was activated for 60 s.

### 2.5. Micro-CT Scanning

Using micro-CT (Nikon Metrology Europe NV, Leuven, Belgium), each tooth was scanned after root canal filling, after re-treatment with Reciproc instruments, and after additional SWEEPS treatment. X-ray parameters (voltage of 110 kV, current of 240 µA) resulted in 26.4 W of X-ray power, which corresponds to a focal spot size of approximately 26 μm. The final CT scan data had a voxel size of 36 μm. A total of 1440 projections with two-frame averaging exposed at 333 ms were taken using a 14-bit flat panel detector. The scanned samples were reconstructed using Volume Graphics VGStudioMax.2 (v3.0, Volume Graphics GmbH, Heidelberg, Germany) with post-processing that included beam-hardening reduction using a Hanning filter, noise reduction using a median filter, and surface detection using an adaptive search algorithm (Volume Graphics VGMax.2). The reconstructed images were processed into three-dimensional volumes using the CT Pro 3D software, Version XT 5.4 (Nikon Metrology Europe NV, Leuven, Belgium) (Figure 1). Analysis of filling remnants, i.e., the presence of gutta-percha in the interior tooth volume, was performed using Volume Graphics VGStudio MAX 3.0 (Volume Graphics GmbH, Heidelberg, Germany). To ensure that results were comparable all inclusion analyses were carried out using the threshold algorithm, with the same parameter of deviation factor and the same probability threshold. In addition, the same surface determination algorithm was used on all samples prior to inclusion analysis. The results indicated a reduction in the volume of root canal filling material on the canal walls after the re-treatment procedure.

### 2.6. Statistical Analysis

Data were statistically analyzed using the SPSS statistical software package (SPSS, v.20, IBM Corp., Armonk, NY, USA) for Windows, with the statistical significance level set to α = 0.05. The results for the volumes of root canal filling after re-treatment with Reciproc instruments and after the additional use of SWEEPS were subjected to linear transformation, to remove the influence of the initial root canal filling volume. The differences between the four experimental groups were tested using an analysis of variance. The achieved sample size was subjected to a power analysis to determine the power of the analysis. The achieved effect size was 0.56 for the re-treatment with Reciproc instruments group and 0.65 for the SWEEPS group, with a statistical significance of *p* < 0.05. The power of the analysis was 0.988 for the re-treatment with Reciproc instruments group and 0.999 for the SWEEPS group. The analysis was performed using G+Power software (University of Kiel, Kiel, Germany).

## 3. Results

In all four experimental groups, the volume of root canal filling materials decreased significantly after re-treatment with the Reciproc instruments (*p* < 0.05), however, complete removal of the root canal filling materials was not observed in any of the teeth (Table 2). There was no statistically significant difference between the groups regarding the volume of root canal filling remnants after re-treatment with Reciproc instruments only (*p* > 0.05) (Table 2). Although SWEEPS significantly reduced the volume of root canal materials in all experimental groups compared with re-treatment with Reciproc instruments alone (*p* < 0.05), even with the SWEEPS treatment none of the teeth showed complete removal of the material from the root canal (Table 2). When additional treatment with SWEEPS was used, a statistically significant difference was found for the single-cone group, with root canals obturated using the combination of AH Plus and gutta-percha, as a higher volume of root canal filling remnants was observed in that group in comparison to the other groups (*p* < 0.001) (Table 2).

## 4. Discussion

The present study tested the efficacy of the SWEEPS technique in the removal of epoxy-resin-based or calcium-silicate-containing sealers in combination with single-cone or carrier-based obturation techniques. The results showed that additional treatment with SWEEPS significantly improved the removal of root canal filling material in all experimental groups, although complete removal was not achieved in any of the samples. A possible explanation for our results is that in the SWEEPS technique, a subsequent laser pulse is transmitted moments before the implosion of the primary bubble, and this action creates a secondary bubble that expands and puts pressure on the primary bubble, hastening its collapse [16,17]. The final result is the emission of primary and secondary shock waves along the entire root canal system [16,17]. This generates a highly dynamic fluid flow that leads to improved chemo-mechanical debridement [17], which might explain the more efficient reduction in the root canal filling in all four experimental groups after using SWEEPS in comparison to conventional re-treatment. This is in accordance with the results of Angerame et al. [18] and Suk et al. [19], who also showed that SWEEPS improved the removal of root canal filling materials. 

The results of the current study also revealed that a combination of bioceramic sealer and bioceramic gutta-percha was more effectively removed from the root canals than the combination of epoxy-resin-based sealer and gutta-percha after SWEEPS treatment of root canals obturated using the single-cone technique. The bioceramic sealer would be expected to be more difficult to remove from the root canals than epoxy-resin-based sealer due to the known adhesion interaction between the dentine walls of the root canal and bioceramic sealers [20]. In this case a mineral infiltration zone is formed, composed of calcium and phosphate ions that undergo a chemical and micromechanical interaction (tag-like structures), which is responsible for the sealing ability and dentine bonding [20]. However, conflicting results have been published regarding the re-treatability of epoxy and calcium silicate sealers. Indeed, some researchers have demonstrated that there are more residual filling materials with bioceramic sealers than with AH Plus during re-treatment with reciprocating instruments [20,21]. Meanwhile, Alsubait et al. [22] found less remaining root canal filling material in canals obturated using calcium silicate sealer, which is consistent with the present study. These results might be related to the penetration depth of the root canal sealers, as one confocal microscopy study revealed a deeper penetration depth for epoxy sealer than for bioceramic sealer [23]. Furthermore, FTIR analysis showed a covalent chemical bond between the epoxide rings of AH Plus and the exposed amino groups of the dentinal collagen [24]. The use of EDTA for the removal of the smear layer, such as in the present study, is beneficial for exposing amino groups of dentin collagen and results in higher push-out bond strengths for AH Plus [24,25]. According to Donnermeyer et al. [26], the push-out bond strength of TotalFill BC sealer was lower in comparison to AH Plus for the single-cone obturation technique, meaning that the micromechanical interaction between root canal dentin and calcium-silicate-based sealers establishes a weaker link to dentin in comparison to epoxy-based root canal sealers. Additionally, EDTA was shown to negatively influence the bond strength of calcium-silicate-based sealers due to the reduction in calcium at the sealer–dentin interface [27]. The strong chemical bond and higher dislocation resistance of AH Plus correlates with its sealing ability and thus its re-treatability, which could explain the results of the present study.

The current study also found a smaller residual volume of root canal filling in the core-carrier group in comparison to the single-cone technique, when AH Plus was used as a sealer, after re-treatment with SWEEPS. This result could be explained by the properties of the AH Plus sealer, which undergoes modification when exposed to heat [28], as occurred in the core-carrier technique used in the present study. AH Plus loses its amine groups upon the application of heat [28], and these groups are setting initiators that are necessary for the polymerization reaction. The result is inadequate sealing of the root canals [28] and, potentially, easier removal of the root canal filling from the root canal, as noted in the current study.

No statistically significant difference was found between the AH Plus and the calcium-silicate-containing sealers in the core-carrier groups regarding the effectiveness of SWEEPS. Previous studies confirmed the lower bond strength of MTA Fillapex to root canal dentin in comparison to AH Plus [29,30,31]. Scanning electronic microscopy demonstrated that, if not exposed to heat, AH Plus exhibited longer, uniform tags, resulting in greater bonding capacity, while MTA Fillapex cement displayed little or no formation of tags [31]. However, the aforementioned modification of AH Plus after exposure to heat [28] might explain the similar removal of both AH Plus and MTA Fillapex from the root canals.

In the two groups in which calcium-silicate-containing sealer was used in combination with different obturation techniques, there was no statistically significant differences in the remaining root canal filling after SWEEPS treatment. These results are in agreement with a recent study that reported no significant effect of the obturation technique on the bond strength of calcium silicate cements, as heat application did not seem to affect the chemical composition of the calcium-silicate–based sealer regardless of the temperature or duration [32]. The authors attributed these findings to the chemical composition of the calcium silicate sealer, which was composed of an inorganic matrix of calcium silicate granules with a water-filled space between them [32]. The heat generated by the core-carrier technique results in water loss in the sealer during obturation [32]. However, the presence of moisture in the dentin tubules may compensate for the loss of water molecules in the setting process. Therefore, the properties of the calcium-silicate-containing sealers would not be influenced by heat [32].

This study has certain limitations, especially concerning comparing the efficacy of SWEPPS in the removal of root canal filling materials with other irrigation techniques; therefore, further studies are needed.

## 5. Conclusions

After an additional SWEEPS treatment, none of the investigated materials were completely removed from the root canal. Nevertheless, SWEEPS improved the removal of both epoxy-resin-based and calcium-silicate-containing sealers in combination with single-cone and carrier-based obturation techniques and can be recommended as an additional procedure in orthograde endodontic re-treatment. SWEEPS was the least efficient in removing the combination of AH Plus and gutta-percha in a single-cone group. 

## Figures and Tables

**Figure 1 materials-16-03273-f001:**
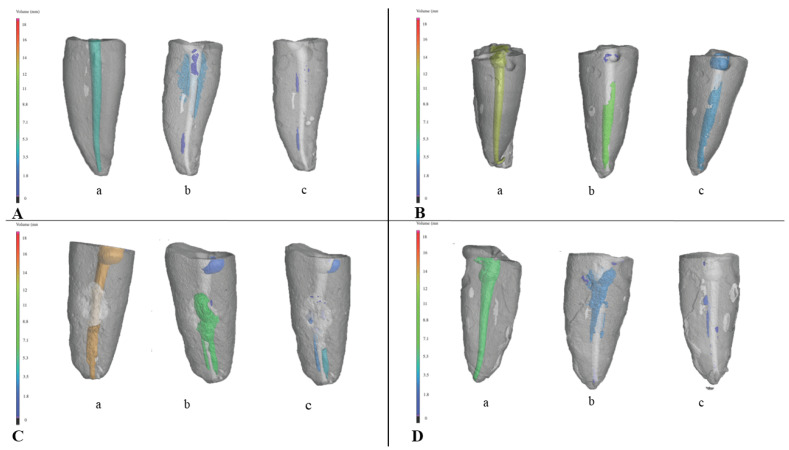
Three-dimensional model of a single canal tooth obtained by micro-CT scanning: (a)—after root canal filling; (b)—after re-treatment with Reciproc instruments; and (c)—after additional SWEEPS treatment. Root canal filling and the root canal filling remnants are colored according to the volume of the root canal filling in: (**A**) Group 1: single-cone group, AH Plus + gutta-percha; (**B**) Group 2: single-cone group, TotalFill BC+ TotalFill BC Points, (**C**) Group 3: core-carrier group AH Plus + Guttafusion; (**D**) Group 4: core-carrier group, MTA Fillapex + Guttafusion.

**Table 1 materials-16-03273-t001:** Composition of root canal sealers.

Root Canal Sealer	Composition
AH Plus^®^	Paste A: bisphenol-A epoxy resin, bisphenol-F epoxy resin, calcium tungstate, zirconium oxide, silica, iron oxide pigments Paste B: dibenzyldiamine, aminoadamantane, tricyclodecane-diamine, calcium tungstate, zirconium oxide, silica, silicone oil
TotalFill BC^™^	Zirconium oxide, calcium silicates, calcium phosphate monobasic, calcium hydroxide, filler and thickening agents
MTA Fillapex^™^	Salicylate resin, diluting resin, natural resin, calcium tungstate, bismuth oxide, nanoparticulate silicate, MTA

**Table 2 materials-16-03273-t002:** Mean values and standard deviation (SD) of the volume (in mm^3^) of root canal filling remnants, determined by micro-CT analysis, in the four experimental groups after root canal re-treatment with Reciproc instruments and additional SWEEPS treatment.

			Reciproc Instruments		SWEEPS	
Groups	Sample Size	Root Canal Filling Technique	Mean	SD	Mean	SD
1. AH Plus + gutta-percha	n = 19	Single-cone	5.0 ^1^	2.1	2.8 ^2^	1.5
2. TotalFill BC + TotalFill BC Points	n = 19	Single-cone	3.5 ^1^	3.3	0.4 ^3^	1.1
3. AH Plus + Guttafusion	n = 19	Core-carrier	3.1 ^1^	1.4	1.0 ^3^	0.8
4. MTA Fillapex + Guttafusion	n = 19	Core-carrier	3.1 ^1^	1.3	0.8 ^3^	0.4
Total			3.7	2.3	1.3	1.4

^1,2,3^ Different number superscripts indicate statistically significant difference (*p* < 0.05).

## Data Availability

The data presented in this study are available on request from the corresponding author.

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
