# Peer review of "Efficacy of Shock Wave-Enhanced Emission Photoacoustic Streaming (SWEEPS) in the Removal of Different Combinations of Sealers Used with Two Obturation Techniques: A Micro-CT Study"

_materials, 2023, doi:10.3390/ma16083273_

Round 1
Reviewer 1 Report
In the present manuscript, the authors approach the topic of the effectiveness of SWEEPS in the removal of some endodontic obturation materials. I appreciate their work and their efforts in achieving their objectives. Still, in order to make their manuscript publishable in the journal, I have some suggestions.
First of all, starting with the title, the word “retreatment” means “a non‐surgical procedure that involves removal of root canal filling materials from the tooth, followed by cleaning, shaping and obturating of the canals”, so the whole procedure. In the present manuscript (not only in the title, but in the whole manuscript), by “retreatment” the authors mean only the removal of the endodontic obturation. So, I suggest to replace the word “retreatment” with “endodontic obturation removal” or something similar.
Some other concerns are related to Materials and Methods section:
Obturation techniques – “All specimens were stored at 37°C for one week to allow sufficient time for the sealer to set” (line 118). In my opinion, the teeth should be kept in a moist environment, at 37°C. If they were, this must be mentioned.
SWEEPS treatment - “The laser was equipped with a dental handpiece (H14; Fotona d.o.o., Ljubljana, Slovenia) and optically coupled with an interchangeable FT” (line 135-136) – what is FT? It’ not mentioned. Please explain
Micro-CT scanning – Please be more explicit about the volume calculation, because almost all of the results are related to the volume modification, but its calculation is not very clear.
Anyway, the major concern is related to the Ethics Committee agreement which is not mentioned in the manuscript. Do the authors have one? Because if not, considering that the study was made on extracted teeth, they should have one. Without it, I’m afraid the manuscript can not be published.
Author Response
On behalf of all authors, I would like to thank the reviewers for the thorough review of the manuscript.Suggestions were constructive and helped us to improve the quality of this manuscript. We agree with all their comments, and we have revised our manuscript accordingly. All changes have been highlighted in yellow color. Please, find our point-to-point response.
Reviewer 1:
COMMENT 1:
First of all, starting with the title, the word “retreatment” means “a non‐surgical procedure that involves removal of root canal filling materials from the tooth, followed by cleaning, shaping and obturating of the canals”, so the whole procedure. In the present manuscript (not only in the title, but in the whole manuscript), by “retreatment” the authors mean only the removal of the endodontic obturation. So, I suggest to replace the word “retreatment” with “endodontic obturation removal” or something similar.
REPLY 1:
The authors have replaced the term “retreatment” with “removal” in the title and throughout the manuscript, referring to the removal of root canal filling materials.
COMMENT 2:
Obturation techniques – “All specimens were stored at 37°C for one week to allow sufficient time for the sealer to set” (line 118). In my opinion, the teeth should be kept in a moist environment, at 37°C. If they were, this must be mentioned.
REPLY 2: Teeth were kept in moist conditions, in saline and this information is now included in the Materials and methods section.
COMMENT 3:
SWEEPS treatment - “The laser was equipped with a dental handpiece (H14; Fotona d.o.o., Ljubljana, Slovenia) and optically coupled with an interchangeable FT” (line 135-136) – what is FT? It’ not mentioned. Please explain
REPLY 3:
FT refers to fiber tips which is now also mentioned in the Materials and methods section.
COMMENT 4:
Micro-CT scanning – Please be more explicit about the volume calculation, because almost all of the results are related to the volume modification, but its calculation is not very clear.
REPLY 4:
As suggested, the volume calculation is now explained in more details in the Materials and methods section.
COMMENT 5:
Anyway, the major concern is related to the Ethics Committee agreement which is not mentioned in the manuscript. Do the authors have one? Because if not, considering that the study was made on extracted teeth, they should have one. Without it, I’m afraid the manuscript can not be published.
REPLAY 5:
Authors confirm the Ethics Committee agreement was obtained at School of Dental Medicine, University of Zagreb and this information is now included in the Materials and methods section.
Reviewer 2 Report
The authors evaluated the effect fo SWEEPS in the retreatment process of different sealers and obturation systems. It becomes clear that SWEEPS enhances the removal of sealer. In general the manuscript is of acceptable quality.
Still, the study has a clear limitation: there is no possibility to compare the results to other studies as a controll group wiht either PUI or EDDY is missing. Such a control group would improve the quality of the study why I would stongly advise to perform additional experiments or at least discuss this limitation.
Author Response
On behalf of all authors, I would like to thank the reviewers for the thorough review of the manuscript.Suggestions were constructive and helped us to improve the quality of this manuscript. We agree with all their comments, and we have revised our manuscript accordingly. All changes have been highlighted in yellow color. Please, find our point-to-point response.
Reviewer 2
COMMENT 1:
The authors evaluated the effect fo SWEEPS in the retreatment process of different sealers and obturation systems. It becomes clear that SWEEPS enhances the removal of sealer. In general the manuscript is of acceptable quality.
Still, the study has a clear limitation: there is no possibility to compare the results to other studies as a controll group wiht either PUI or EDDY is missing. Such a control group would improve the quality of the study why I would stongly advise to perform additional experiments or at least discuss this limitation.
REPLAY 1:
Authors do acknowledge that, like every in vitro study, this one as well has certain limitations. In our study, the control group was conventional irrigation only. Our plan is to conduct further investigation in the future comparing SWEEPS to other activated irrigation techniques. However, this limitation of our study is now mentioned in the manuscript.
Reviewer 3 Report
Manuscript ID: materials-2310947
Title: Efficacy of shock wave-enhanced emission photoacoustic streaming (SWEEPS) in the retreatment of different combinations of sealers and obturation techniques: A micro CT study
Thank you for giving me the opportunity to review this manuscript. It is a well-designed study and well written manuscript which is of publishable standard. However, some minor clarifications and suggestions would improve its readability and make it more robust.
Abstract:
Comment: The abstract is very well written and adequately describes and represents the manuscript.
Introduction:
This is also adequately written with appropriate review of the relevant literature. However, it is suggested that it would be helpful before the objectives (that is clearly outlined)is to include a research question and hypothesis or a null hypothesis. Please edit accordingly.
Materials and Methods:
It reads on pages 2 and 3, lines 95 to 97: “The canals were obturated using either a cold single-cone technique (groups 1 and 2) or a warm core-carrier technique (groups 3 and 4).
Comment: Who carried out the Root canal preparation and obturation and who made the retreatment process? Weather it is one person or several, please clarify if calibration and reliability processes were made. Please clarify and edit accordingly.
Comment: On page 3 lines 98 to 116, it details the combination of obscuration and sealant for each group. Please clarify what is the rational of using these specific combinations.
It reads on page 3, lines 122: “After one week and complete setting of the sealer, the root canal retreatments were 122 performed on all specimens (n=76).”
Comment: Please clarify the reason for the timing of one week. Is there a specific scientific rational for doing the retreatment after one week? Please clarify and edit accordingly.
Results:
It reads on page 5, lines 190 to 194 “Statistical analysis of the effectiveness of SWEEPS according to the root canal filling technique or the root canal filling materials revealed a statistically significant difference for the single-cone group treated with the combination of AH Plus and gutta-percha, as a higher volume of root canal filling remnants was observed in that group than in the other groups (p<0.001) (Table 2).
Comment: Please add a clinical significance to this sentence.
Both Discussion and Conclusions are appropriate
End Report
Author Response
On behalf of all authors, I would like to thank the reviewers for the thorough review of the manuscript.Suggestions were constructive and helped us to improve the quality of this manuscript. We agree with all their comments, and we have revised our manuscript accordingly. All changes have been highlighted in yellow color. Please, find our point-to-point response.
COMMENT 1:
Introduction:
This is also adequately written with appropriate review of the relevant literature. However, it is suggested that it would be helpful before the objectives (that is clearly outlined)is to include a research question and hypothesis or a null hypothesis. Please edit accordingly.
REPLY 1:
Null hypothesis was added in the Introduction section as suggested.
COMMENT 2:
Materials and Methods:
It reads on pages 2 and 3, lines 95 to 97: “The canals were obturated using either a cold single-cone technique (groups 1 and 2) or a warm core-carrier technique (groups 3 and 4). Comment: Who carried out the Root canal preparation and obturation and who made the retreatment process? Weather it is one person or several, please clarify if calibration and reliability processes were made. Please clarify and edit accordingly.
REPLY 2:
In the materials and methods section, requested details were added.
COMMENT 3:
Comment: On page 3 lines 98 to 116, it details the combination of obscuration and sealant for each group. Please clarify what is the rational of using these specific combinations.
REPLY 3:
The idea of the study was to compare the efficacy of SWEEPS in removal of more recent type of root canal sealers, calcium silicate based sealers in comparison to epoxy resin based sealer, as a control group, but when combined with two obturation techniques, single cone, often used nowdays after instrumentation using engine driven instruments and one warm obturation technique (carrier-based) which is also often used, especially in cases when cold techniques can not ensure sealing of the endodontic space.
COMMENT 4:
It reads on page 3, lines 122: “After one week and complete setting of the sealer, the root canal retreatments were 122 performed on all specimens (n=76).” Comment: Please clarify the reason for the timing of one week. Is there a specific scientific rational for doing the retreatment after one week? Please clarify and edit accordingly.
REPLY 4:
This one week period is always indicated in similar studies as it was shown that, in vivo, it can be expected that the root canals sealers have completely set so this is the reason for choosing this same period frame before retretmant (e.g. Silva EJNL, Ehrhardt IC, Sampaio GC, Cardoso ML, Oliveira DDS, Uzeda MJ, Calasans-Maia MD, Cavalcante DM, Zuolo ML, De-Deus G. Determining the setting of root canal sealers using an in vivo animal experimental model. Clin Oral Investig. 2021 Apr;25(4):1899-1906. doi: 10.1007/s00784-020-03496-x. Epub 2020 Aug 13. PMID: 32789655.).
COMMENT 5:
Results:It reads on page 5, lines 190 to 194 “Statistical analysis of the effectiveness of SWEEPS according to the root canal filling technique or the root canal filling materials revealed a statistically significant difference for the single-cone group treated with the combination of AH Plus and gutta-percha, as a higher volume of root canal filling remnants was observed in that group than in the other groups (p<0.001) (Table 2). Comment: Please add a clinical significance to this sentence.
REPLY 5:
Clinical significance of this sentence was added in the Conclusion section of the manuscript.
Round 2
Reviewer 1 Report
The authors made all the suggested recommendations, so, I consider the manuscript can now be published.
Author Response
All authors would like to thank all reviewers for the comments and effort.
Reviewer 2 Report
Thank you for providing a revised version of the manuscript. I should be suitable for publication now.
Author Response

(The authors gave the same response as above.)
